# DISENTANGLING ONE FACTOR AT A TIME

## ABSTRACT

With the overabundance of data for machines to process in the current state of machine learning, data discovery, organization, and interpretation of the data becomes a critical need. Specifically of need are unsupervised methods that do not require laborious labeling by human observers. One promising approach to this enedeavour is *Disentanglement*, which aims at learning the underlying generative latent factors of the data. The factors should also be as human interpretable as possible for the purposes of data discovery. *Unsupervised disentanglement* is a particularly difficult open subset of the problem, which asks the network to learn on its own the generative factors without any link to the true labels. This problem area is currently dominated by two approaches: Variational Autoencoder and Generative Adversarial Network approaches. While GANs have good performance, they suffer from difficulty in training and mode collapse, and while VAEs are stable to train, they do not perform as well as GANs in terms of interpretability. In current state of the art versions of these approaches, the networks require the user to specify the number of factors that we expect to find in the data. This limitation prevents "true" disentanglement, in the sense that learning how many factors is actually one of the tasks we wish the network to solve. In this work we propose a novel network for unsupervised disentanglement that combines the stable training of the VAE with the interpretability offered by GANs without the training instabilities. We aim to disentangle interpretable latent factors "one at a time", or OAT factor learning, making no prior assumptions about the number or distribution of factors, in a completely unsupervised manner. We demonstrate its quantitative and qualitative effectiveness by evaluating the latent representations learned on three benchmark datasets; DSprites, 3DShapes and CelebA.

## 1 INTRODUCTION

Deep learning models, which are now widely adopted across multiple A.I. tasks ranging from vision to music generation to game playing (Krizhevsky et al., 2017; Oord et al., 2016; Mnih et al., 2015), owe their success to their ability to learn representations from the data rather than requiring hand-crafted features that older models required. However, this self-learning of abstract representations comes at the known cost of the resulting representations being cryptic and inscrutable to human observers. These learned representations might be dramatically affected by noise or spurious correlations between the data and the labels - the representations might encode 'useless' information from the input data which is correlated with the corresponding label Geirhos et al. (2020). This makes them more vulnerable to slight changes in the data distribution. A more comprehensive understanding of the data down to essential indivisible factors would allow us to learn insights, sort and label data, and facilitate downstream learning more efficiently. This also requires, critically, that these factors be somehow interpretable as well. This approach, dubbed *Disentanglement*. requires that we learn the data from its fundamental building block, so-called "disentangling" the true factors - the *factors of variation* - that generate the data. If one were to learn these factors, one would learn all possible causes of variation in a given dataset, and would in some sense gain complete understanding of the underlying machinery, so to speak. In this work, we propose a new method that attempts to learn the true disentangled factors one at a time, in a way that maintains interpretability for future use.

---

[0]Code available at https://github.com/OATFactor/OATFactor

## 1.1 What is a disentangled representation?

While it is easy to informally talk of factors of variation, actually pinning down concrete definitions of disentangled learning has proved a somewhat more difficult task. Though there is no commonly accepted formalized notion of disentanglement or validation metrics (Higgins et al., 2018), recent works have characterized disentangled representations, based on natural intuition, as one which encodes each informative factor of variation of the data in separate latent dimenions (Bengio, 2013), such that a change in single factor of variation produces a change in only a subset of the learned latent representation. This is refered to as the *separability* (Do & Tran, 2020) quality of the representations, also called disentanglement (Eastwood & Williams, 2018) and modularity (Ridgeway & Mozer, 2018). Separability ensures that the downstream tasks which depend on a certain subset of factors are not affected by changes in other factors thus facilitating robust models Suter et al. (2019).

Some previous works (Ridgeway & Mozer, 2018; Eastwood & Williams, 2018) have suggested that a single dimension of the learned disentangled representation should completely describe a factor. This constraint along with separability entails a bijective mapping between the true factors of variation and the learned representation. However, enforcing this constraint might not be conducive to learn complex factors in a single latent variable Esmaeili et al. (2018). Hence in this work we focus primarily on separability. For an in depth discussion on interpretability and how we enforce it in the proposed work, we refer the reader to A.1.

## 1.2 What is missing from current approaches to solving disentanglement?

Due to the monumentous task of learning data without supervision, current methods have attempted to tackle the subproblems of separability and interpretability, but few have successfully solved both at once. Learning these disentangled representations in a *semi-supervised* setting Kulkarni et al. (2015); Siddharth et al. (2017); Locatello et al. (2019) is a relatively easier task where additional annotated data is available that give a strong backpropgation gradient to the network to guide it to cleanly separate factors. However, if the main goal of disentanglement is to discover unknown factors in large data corpuses, then it is not the right direction to require labels; rather the network should learn in an unsupervised fashion. For unstructured data, all current state of the art approaches based on *unsupervised* disentanglement rely on a deep generative neural network, built either on a Variational Autoencoder or Generative Adversarial Network structure (See Related Work.) Many popular VAE based methods are able to do well in separating factors out but make strict assumptions on the number of factors and their structure and those that do, do not explicitly attempt to also make the factors interpretable. Other methods based on GANs suffer from common issues from training of GANs, but also tend to only learn a small subset of factors (See 2). Here we propose a novel *hybrid* network that provides the stability and strong gradients of VAE training with the performance of GAN training, without mode collapse.

Most current VAE-based state of the art methods make the assumption that there are a fixed number of independent factors for all the data points in the dataset. However in real datasets, in addition to the independent factors common to all points in the dataset, there might also be some correlated, nuisance or noisy factors pertinent to specifically only certain data points. Moreover, approaches rely on a heuristically chosen latent dimension d, sufficiently large to encompass all true factors. However, this suffers from the same pitfalls for many of the same reasons that algorithms like k-means cluster do, namely that given a new, unseen dataset, we do not necessarily know how many independent factors there are. In fact, this is one of the main goals of disentanglement in the first place, to glean insights about the data. Our method instead assumes that there is a set of independent factors, and one of entangled nuisance and correlated factors, and separates them out. We then iteratively learn to disentangle each factor of variation one at a time, such that the network learns on its own different independently controllable factors thus removing a current hand-tuned roadblock on the way to true unsupervised disentanglement. Second, in the same training loop, we ensure that the disentangled latent representation follows *intepretable latent code manipulation* (Section. 1.1), which says that a change in a single latent should make a distinct and noticeable change in the output (Eastwood & Williams, 2018). Together, we demonstrate our proposed model is able to learn both critical qualities of disentanglement, in a completely unsupervised manner.

### 1.3 CONTRIBUTIONS

Our main contributions are as follows:

- We introduce a new completely unsupervised generative neural model, One at a Time (OAT) factor learning that combines the stability of VAE training with the accuracy of GAN learning, which contains both a set of independent separable factors and a set of entangled factors, that produces separable and interpretable latent factor codes

- Our proposed model is the first unsupervised method that is capable of learning an arbitrary number of latent factors via incremental unsupervised interventions in the latent space

- We test and evaluate our algorithm on three datasets and across multiple metrics. Our empirical results strongly suggest that our proposed method is effective in finding interpretable factors and competitive with the most recent disentanglement strategies

## 2 RELATED WORK

Various authors have attempted to learn unsupervised disentangled representations using generative models in recent years. State-of-the-art approaches for unsupervised disentanglement learning can be broadly classified into two categories based on the type of generative model used; one via Variational Autoencoders (VAE) (Kingma & Welling, 2014; Rezende et al., 2014), and another via Generative Adversarial Networks (GAN) (Goodfellow et al., 2014).

### 2.1 VIA VARIATIONAL AUTOENCODERS

Variational Autoencoders are a deep generative neural network model which learns an approximate posterior distribution of the latent representations from the data, while trying to maximize the data log-likelihood. They can be thought of as an autoencoder with an additional loss term that drives the reconstructions to be closer together in latent space. VAEs assume that the data $x$ is generated from a set of latent features $z$ with a prior $p(z)$ according to the model $p_\theta(x|z)p(z)$. The top-down generator $p_\theta(x|z)$ and a bottom up inference network $q_\phi(z|x)$ are modeled as multilayer neural networks and trained jointly to maximize the marginal log-likelihood of the empirical distribution of the training data. However, since the marginal log-likelihood is intractable, VAEs optimize a a tractable lower bound $\mathcal{L}$ on the data log-likelihood $p_\theta(x)$ called the Evidence Lower Bound (ELBO):

$$\mathcal{L} = \mathbb{E}_{q(x)}[\mathbb{E}_{q_\phi(z|x_i)}[\log p_\theta(x_i|z)] - \mathrm{KL}(q_\phi(z|x_i)||p(z))] \tag{1}$$

where $q(x) = \frac{1}{N}\sum_{i=1}^{N}\delta(x_i)$ is the distribution of the training data. The first term measures the reconstruction error and the second term measures the distance between the approximate posterior distribution $q_\phi(z|x)$ and the assumed prior distribution $p(z)$. Many state-of-the-art unsupervised disentanglement methods extend the above objective function to impose additional constraints on the structure of the latent space to match the independent prior assumption. $\beta$-VAE (Higgins et al., 2017) and AnnealedVAE (Burgess et al., 2018) heavily penalize the KL divergence term thus forcing the learned posterior distribution $q_\phi(z|x)$ to be independent like the prior. Factor-VAE (Kim & Mnih, 2019) and $\beta$-TCVAE (Chen et al., 2019) penalize the total correlation of the aggregated posterior $q_\phi(z)$. $TC = KL(q(z)||\prod_{i=1}^{K}q(z_i))$ where the aggregated posterior is calculated as $q_\phi(z) = \mathbb{E}_{p(x)}[q(z|x_i)] = \frac{1}{N}\sum_{i=1}^{N}q_\phi(z|x_i)$ using adversarial and statistical techniques respectively. DIP-VAE (Kumar et al., 2018) forces the covariance matrix of the aggregated posterior $q(z)$ to be close to the indentity matrix by method of moment matching. Other works improved the performance by making a specific design for discrete factors (Dupont, 2018; Jeong & Song, 2019); and use optimization techniques based on annealing to encode information effectively in the discrete and continuous factors.

### 2.2 VIA GENERATIVE ADVERSARIAL NETWORKS

Models based on GANs, explicitly condition the generator network with a set of independent latent variables $c$ (by concatenation with random noise $z$), and train the generator to generate data which has high mutual information with $c$. The most prominent work from the GAN family is InfoGAN

(Chen et al., 2016) which learns disentangled, semantically meaningful representations by maximizing a lower bound on the intractable mutual information between the conditioning latent variables $c$ and the generated samples $G(z, c)$.

$$\min_G \max_D \mathcal{L}(D, G) - \lambda I(c; G(z, c)) \tag{2}$$

where the adversarial loss is given by;

$$\mathcal{L}(D, G) = \mathbb{E}_{x \sim P_x}[\log(D(x))] + \mathbb{E}_{z \sim p(z), c \sim p(c)}[\log(1 - D(G(z, c)))] \tag{3}$$

Assuming a perfect discriminator, the generator tries to minimize the Jensen-Shannon Divergence between the true data distributing and the generated distribution. By changing the value of the conditioning variables, the generator is forced to make distinct and noticeable changes in the data, such that the value of the conditioning variables can be recovered from the generated data alone. InfoGAN-CR (Lin et al., 2020) add a contrastive regularizer to the InfoGAN model, which is trained to predict the changes in the latent space given only the pairs of images generated from the respective latent codes. (Zhu et al., 2020) augment their objective with a similar self supervised learning task to predict the dimension of the latent variable which is different from the pair of images. Some other works based on GANs are (Jeon et al., 2019; Liu et al., 2020). (Liu et al., 2020) add orthogonal regularization to encourage independent representations.

Alternatively, approaches based on the InfoGAN framework find interpretable factors of variation through the Information Maximization principle (InfoMAX). However, in many cases these approaches suffer from mode collapse - a phenomenon that causes complete failure in training. Critically, GANs lack the ability to provide an *explicit* modeling of the latent space, as VAEs do (by explicitly learning the parameters of an encoder distribution), as the entire GAN model relies on implicit sampling. We aim to address this issue by our hybrid approach combining VAEs with GAN components.

Suter et al. (2019) introduced the concept of interventions to study the robustness of the learned representations under the Independent Mechanisms (IM) (Schoelkopf et al., 2012) assumption. In this paper, however, we use the method of interventions while training the model to find disentangle common factors from their entangled set. (Lee et al., 2020) use a VAE to disentangle and then pass into a GAN to generate high-fidelity images; our approach differs in that the GAN component is tightly integrated into our training loop, we perform interventions, and we split the latents into two spaces (See Sec. 3). Hu et al. (2018) condition the latent space of entangled representations with a disentangled code, which they learn in a supervised way using specific attribute discriminators. As far as the authors are aware, we are the first to split the latent dimension into entangled and disentangled in a completely unsupervised way, as well as the first to combine this with interventions, and with incremental learning.

## 3 OUR METHOD

### 3.1 GENERATIVE DISENTANGLEMENT

Most previous works on disentanglement take a *generative* view of data where they assume that an unknown generative model has produced the data. The data is assumed to be composed from an *a-priori* set of *factors of variation* $G_k (k = 1, \cdots, K')$, which contains the different human-defined atomic features that assume different values for specific instances. Here, $K'$ is assumed to be the "true" number of independent factors of variation in the data. The data is then assumed to be generated by a two-step process. First, the values of the different factors of variation, $g$, are sampled from a factorized distribution $p(G) = \prod_{k=1}^{K'} p(G_k)$, where $p$ can be any probability distribution. A *generator function* $p(x|G)$ then maps the specific values, $g$, sampled from $p(G)$, to a high-dimensional datapoint $x$. Thus, $p(x|g)$ describes a causal mechanism (Suter et al., 2019) invariant to changes in the distributions $p(g_i)$. In this generative view the aim of disentangled representation learning then becomes to uncover these "true" factors of variation from the data alone and re-encode each factor as an independent latent representation.

We posit, however, that this generative view of disentanglement is too limiting and restrictive when it comes to real-world data, as described in section 1.1. Even though the factors are independent concepts, knowing what observation of $x$ we obtained renders the different latent causes dependent

as certain factor realizations tend to co-occur more than others as a characteristic of the dataset. In the proposed work we do not assume any particular factorization of the distribution of the factors of variation and instead separate out the factors through iterative interventions from their entangled counterpart as discussed in further sections. These interventions ensure that the dimensions of the latent representation are independently controllable and have no causal effect on each other. Moreover, independent interventions also ensure that the confounding variables for the different factors are integrated out. In addition to the generative factors of variation of the dataset, we also model a second set of entangled, correlated nuisance factors pertinent to that particular data point. We train our network to systematically discern the meaningful latent factors shared across the dataset from the nuisance ones, both of which are important to maximize the log-likelihood of the data (See 3.3 for details).

## 3.2 DISJOINT ENTANGLED AND DISENTANGLED LATENT SETS

In practise training a VAE, guided by the reconstruction loss to maximize the log-likelihood of the data does not lead to disentangled representations, i.e., do not adequately separate out all of the factors within one latent space $z$. While constraints can be imposed on the latent space to enforce independence between the different dimensions of the learned representations, they can lead to blurry reconstructions (Kim & Mnih, 2019) where some factors might be ignored altogether. Because of these practical limitations, we propose a novel method of splitting the latent space $\mathbf{z}$ into two disjoint sets: the disentangled set $\mathbf{z}_1 = \{z_1^1, z_1^2, \cdots, z_1^K\}$ where $z_1^k \in \mathbb{R}$ and the entangled set $\mathbf{z}_2 \in \mathbb{R}^d$. Here, $K$ is the number of factors that the network learns, and is not predetermined but upper-bounded. This factorizes the generative model as: $p_\theta(x, z) = p_\theta(x|z_1, z_2)p(z_2) \prod_i^K p(z_1^i)$.

During training, the factors common to the dataset are disentangled into $\mathbf{z}_1$, while the nuisance factors that are either correlated with the other factors in the dataset, or are noise factors that are specific to individual samples, but are not representative of the dataset as a whole are encoded in $z_2$ which has a higher encoding capacity. This ensures that the learning of the disentangled latent set is not solely driven by the reconstruction cost. Our proposed model is the first VAE-based model to incorporate this notion of two sets of latent variables, which were used in a somewhat similar manner in GANs in (Chen et al., 2016; Lin et al., 2020).

## 3.3 PROPOSED ARCHITECTURE

Our proposed model consists of an encoder and a decoder as in a standard VAE, modeled as deep convolutional neural networks, with an additional discriminator network attached (see Fig. 1). The encoder and decoder network parameterize the posterior distribution of the latent representation $q_\phi(\mathbf{z}|x)$ and the generative model distribution $p_\theta(x|\mathbf{z})$ respectively. In addition to the standard VAE, we attach a discriminator network $D_w$ to the output of the VAE decoder. This ensures that the distribution of images generated by the intervention procedure (See Sec 3.4.2) is close to the distribution of the images in the training set and thus changes in the latent space translate only to interpretable changes. This discriminator distinguishes between the true images in the dataset and generated images by the decoder by using the "real or fake" paradigm of GAN training.

Per the discussion in Section 3.2, the latent representation layer is divided into two sets, a disentangled set $\mathbf{z}_1$ and a correlated entangled set $\mathbf{z}_2$. Thus the entire latent space is denoted as $\mathbf{z} = (\mathbf{z}_1, \mathbf{z}_2)$. We use the boldface $\mathbf{z}$ to represent a vector-valued random variable. The encoder network encodes the data into the two sets of latent variables $q_{\phi_1}(\mathbf{z}_1|x)$ and $q_{\phi_2}(\mathbf{z}_2|x)$, with $\phi_1$ and $\phi_2$ sharing weights for a set number of initial layers. The two sets are then passed through the decoder $p_\theta(x|\mathbf{z})$ to yield the reconstructed images $\hat{x}$.

## 3.4 TRAINING PROCEDURE

Our training procedure consists of a pre-training phase and a main learning procedure: a pre-training phase where we train a VAE to maximize the data log-likelihood, and the main training phase, where the OAT factor learning is performed.

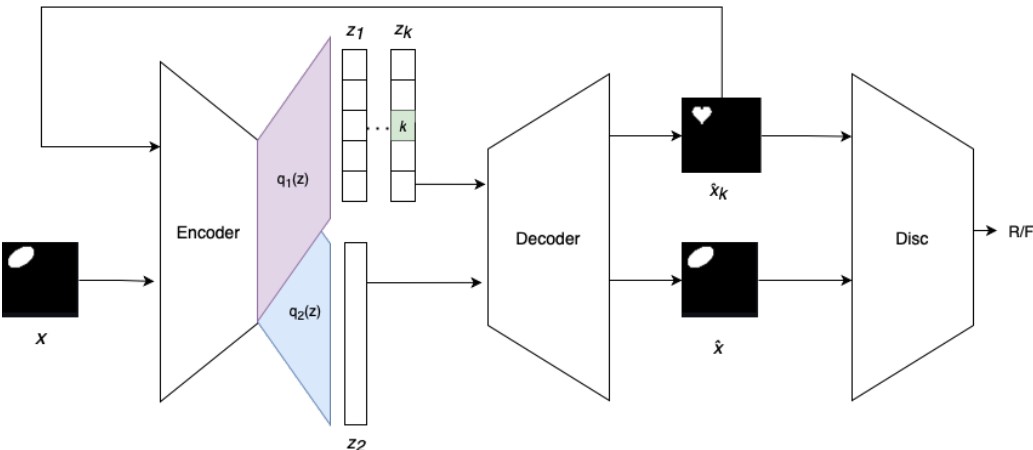

Figure 1: The complete OAT architecture. First, an input image, $x$, is passed into the VAE encoder, a deep convolutional neural network (CNN), and encoded via two multi-layered components $q_{\phi_1}(z_1|x)$ and $q_{\phi_2}(z_2|x)$, into two distinct latent spaces, a "factorized" or disentangled space, $z_1$, and a correlated space, $z_2$, which is then decoded by a deep transpose convolutional neural network, to produce a reconstructed image $\hat{x}$. The insight of OAT training is that it may not be possible to decorrelate all of the data for various reasons, so we first group the correlated latents into one space, $z_2$, and then "peel off" each independent factor one at a time. Next, an *intervention* is made on one latent variable in the new disentangled space, $z_1^1$, creating a new latent $\mathbf{z}^1$, which is passed through the decoder to produce a new image $\hat{x}^1$. This factor-reconstructed $\hat{x}^1$ is then passed back through the encoder to ensure the encoder learns how to encode that particular factor change into the same intervention-altered disentangled latent $\mathbf{z}^1$. The factor-reconstructed values $\hat{x}^1$ are then passed into a discriminator $D_w$ along with real images $x$, to ensure that the factor-altered reconstructions remain realistic.

### 3.4.1 VAE PRE-TRAINING

We first train the encoder and the decoder to maximize the likelihood of the training data under the generative model $p_\theta(x \,|\, \mathbf{z}_2)$ as detailed in equation 1. In this step, only the $\mathbf{z}_2$ latent set is learned; which encodes all the informative factors of variation albeit in an entangled way. Since we do not enforce any extra independence constraints on the latent space, the learning of the latent space is driven by the reconstruction loss. The posterior distribution $q_{\phi_2}(\mathbf{z}_2|x)$ is regularized to be similar to the zero mean, unit variance, isotropic Gaussian prior $p(\mathbf{z}_2)$. Thus, the objective function we aim to minimize is evidence lower-bound (ELBO) of the data log-likelihood as in a regular VAE.

$$\mathcal{L}_{\text{ELBO}} = \mathbb{E}_{q_{(x)}}[\mathbb{E}_{q_{\phi_2}(\mathbf{z}_2|x)}[p_\theta(x|\mathbf{z}_2)] - \text{KL}(q_{\phi_2}(\mathbf{z}_2|x)||p(\mathbf{z}_2))] \tag{4}$$

where $p(\mathbf{z}_2) = \mathcal{N}(0, I_d)$

The first term in the above objective function minimizes the reconstruction error of the data points from the latent representations alone. This ensures that the latent representations encode the important information in the data or all the different factors of variation in the dataset.

### 3.4.2 OAT FACTOR DISENTANGLING

The main contribution of our work occurs at the this stage, where we perform OAT factor disentanglement. We outline the process as a two-step process:

**Step 1: Passing Through The Disentangled Latents**

Once the pre-training is completed, the reconstruction loss saturates and all the informative factors are encoded in $\mathbf{z}_2$, however they are highly entangled. In step 1, we perform the same VAE training pass as in the pre-training phase, but we now pass the data also through $\mathbf{z}_1^{1:k}$, where $k$ is the number of dimensions learned until that point. This will eventually allow the model to encode information in $\mathbf{z}_1$ from the data in a disentangled way instead of their entangled representations in $\mathbf{z}_2$. For brevity

we denote the set $(\mathbf{z}_1^{1:k}, \mathbf{z}_2)$ as $\mathbf{z}$. Thus objective function for this step is as follows:

$$\mathcal{L}_1 = \mathbb{E}_{q_{(x)}}[\mathbb{E}_{q_\phi(\mathbf{z}|x)}[p_\theta(x|\mathbf{z})] - \beta \text{KL}(q_{\phi_2}(\mathbf{z}_2|x)||p(\mathbf{z}_2)) - \sum_{i=1}^{K} \gamma_i \text{KL}(q_{\phi_1}(z_1^i|x)||p(z_1^i))] \quad (5)$$

where $p(z_1) = \prod_{i=1}^{k} p(z_1^i) = \prod_{i=1}^{k} \mathcal{N}(0,1)$. Here $\gamma_i$ is a weighting factor that acts as a mask, which indicates turning off latent elements of $z_1$ that are not being currently trained as part of the OAT iterative procedure. During the pre-training $\gamma_i\{i = 1 : K\}$ are set to zero. At the beginning of the OAT training procedure, only the first element of $z_1$, $z_1^1$, is learned, with $\gamma_1 = 1$ and for the latents with $\gamma_i\{i = 2 : k\} = 0$ the gradients are not calculated and the weights are not updated. This constitutes a core of the "one at a time" component, and is critical for the network's functionality: by focusing on one latent factor at a time, we can iteratively learn and discover each latent factor at different points during the training. As the training proceeds the value of more $\gamma$'s are flipped to 1 which allows more latents in $\mathbf{z}_1$ to be seen by the network.

The value of $\beta$ is increased linearly during training to encourage the model to encode information in $\mathbf{z}_1$ instead of $\mathbf{z}_2$. This along with the intervention described in the further sections ensure that the posterior distribution of the disentangled set does not collapse to the prior.

**Step 2: Interventions and Change-Discriminator**: In order to ensure that an independent, interpretable factor of variation is encoded in $z_1^i$, we perform *interventions* on each of the learned dimensions of $z_1$. Thus it is important for any particular dimension $i$ in $z_1$, that changing the value of another dimension $z_1^j (j \neq i)$ should not change the value of $z_1^i$ when the corresponding generated data from the intervention is re-encoded. ***Interventions:*** For the intervention procedure, we start with a learned representation $\mathbf{z} = \{\mathbf{z}_1, \mathbf{z}_2\}$ such that $\mathbf{z}_1 \sim q_{\phi_1}(\mathbf{z}_1|x)$ and $\mathbf{z}_2 \sim q_{\phi_2}(\mathbf{z}_2|x)$, encoded from an some datapoint $x$. We then uniformly select a dimension $k \in [K]$ from the learned dimensions of $\mathbf{z}_1$ to change it's value during intervention. We sample a new value for the dimension $k$ from the prior distribution $p(z_1^k)$ say $c$ to create a new representation $\mathbf{z}^k = (\{c, z_1^{\backslash k}\}, \mathbf{z}_2)$. Thus the two representations are same in all dimensions except $k$. The new representation is then passed through the decoder to generate a data point $\hat{x}^k$. This generated data is then re-passed back through the encoder to obtain the representations $\hat{\mathbf{z}}^k = (\{\hat{c}, \hat{z}_1^{\backslash k}\}, \hat{\mathbf{z}}_2)$ sampled from the encoder distributions $q_{\phi_1}(\mathbf{z}_1|\hat{x}^k)$ and $q_{\phi_2}(\mathbf{z}_2|\hat{x}^k)$ respectively. Both the encoder and the decoder are then trained to reconstruct $\mathbf{z}^k$ according to 6. We refer to this procedure of altering a single factor, combined with data generation and re-passing through the encoder, an *intervention*, per Suter et al. (2019).

While we want to reconstruct the intervened latent variable, we also want to ensure that the changes, the intervened latent variable makes in the corresponding generated data, are interpretable. This is ensured by constraining the distribution of the generated data to lie strictly in the true data manifold. Thus a particular generated datum differs from the training datum in a subset of factors of variation, ideally in only one factor corresponding to the intervened latent variable. To address this, we train the discriminator $D_w$ to differentiate between the training data and the generated data. The decoder weights are updated to generate data which would fool the discriminator. Thus the complete objective function can be written as follows;

$$\mathcal{L}_2 = \mathbb{E}_{p(z)}[\mathbb{E}_{p_\theta(\hat{x}^k|\mathbf{z}^k)}[q_\phi(\mathbf{z}^k|\hat{x}^k)] - \beta \text{KL}(p_\theta(\hat{x}^k|\mathbf{z}^k)||q(x))] \quad (6)$$

The first term in the above objective function minimizes the reconstruction error of the re-encoded, intervened latent representation from the generated data. Because the intervened representations are reconstructed from the generated data alone, the decoder is forced to make distinct changes in the generated images corresponding to the different dimensions of $\mathbf{z}_1$. This ensures that the disentangled set $\mathbf{z}_1$ is independently controllable and interpretable, a desirable property of disentangled representations.

Since it is difficult to compute analytically the high-dimensional true data distribution from the samples alone, the KL divergence in the second term is replaced with the Jensen-Shannon Divergence (JS-divergence). In Goodfellow et al. (2014) given an optimal discriminator network $D_w$, described in Sec. 3.3, training the decoder to fool the discriminator essentially minimizes the JS-divergence between the generated distribution and the training images.

During the training, step 1 and step 2 are performed one after the other in an alternative fashion, similar to other two-step learning algorithms such as wake-sleep (Hinton et al., 2006). This process

| Model | FactorVae | MIG | DCI | BetaVAE |
|---|---|---|---|---|
| VAE | 0.63±.06 | 0.10 | 0.30±.10 | |
| $\beta$-VAE | 0.63±.10 | 0.21 | 0.41±.11 | |
| FactorVAE ($\gamma$=40) | 0.82±.01 | 0.43±.01 | 0.74±.01 | 0.84±.01 |
| $\beta$-TCVAE | 0.62±.07 | **0.45** | 0.29±.01 | |
| InfoGAN | 0.82±.01 | 0.22±.01 | 0.60±.02 | 0.87±.01 |
| InfoGAN-CR | **0.88±.01** | 0.37±.01 | 0.71±.01 | **0.95±.01** |
| OAT ($z_1$=10, $z_2$=10) | 0.82 ± .11 | 0.36 ± .13 | **0.78± .01** | 0.80 ± .11 |
| OAT ($z_1$= 5, $z_2$=10) | 0.84 ± .12 | 0.44 ± .03 | 0.74± .05 | 0.82 ± .10 |
| OAT ($z_1$=12, $z_2$=10) | 0.78 ± .08 | 0.33 ± .11 | 0.77± .01 | 0.76 ± .08 |

Table 1: Comparisons of the popular disentanglement metrics on the dSprites dataset. A perfect disentanglement corresponds to 1.0 scores. The scores are averages over 10 runs with different random seeds.

has the effect of minimizing the symmetric KL divergence between the joint generative and inference distributions $p_\theta(x, z)$ and $q_\phi(x, z)$ A.

During training, the first latent variable of the set $z_1^1$ is trained and thus $\gamma_1$'s value is changed from 0 to 1. This latent is trained using interventions until it's KL-divergence saturates and it can encode no more information. Then, we start training the next latent variable from the set $z_1$ i.e. $z_1^2$ and the value of $\gamma_2$ is switched from 0 to 1. The stopping criteria for this process depends on the dataset. If the dataset has a fixed number of factors which are independent like in the synthetic datasets dSprites and 3DShapes, the process of adding more latents is stopped when the KL-divergence of the entangled set, $z_2$, goes to zero and it encodes no more information. That is when we can say that all the factors of variation have been found. For real datasets like CelebA, we can find continue finding factors until the desired number of factors have been found.

## 4 EMPIRICAL EVALUATION

For quantitative evaluation, we run experiments on two synthetic datasets generated from independent ground truth factors of variation; dSprites Matthey et al. (2017) and 3DShapes Burgess & Kim (2018). For qualitative evaluation we use a real dataset with unknown factors of variation, the CelebA dataset Liu et al. (2015).

We evaluate the learned representations with one metric from each of the three kinds of metrics as described in (Zaidi et al., 2021). We use the FactorVAE (Kim & Mnih, 2019), the BetaVAE(Higgins et al., 2017), Mutual Information Gap (MIG) (Chen et al., 2019) and the Disentanglement-Completeness-Informativeness (DCI) (Eastwood & Williams, 2018) metric to quantitatively evaluate disentanglement on dSprites (1) and 3DShapes (2). We also qualitatively evaluate the models trained on CelebA 4, dSprites 4, 3D shapes (ref: A.4). More details regarding the implementation and the metrics and the latent traversals can be found in the appendix **??**.

We also perform ablation studies on the dimension of $\mathbf{z}_1$, K, and $\mathbf{z}_2$, d, on the dSprites dataset. We find that increasing d beyond a certain range (5-10), does not affect the metrics but make the reconstructions sharper as shown in A.4. The value of K affects how the factors that are encoded in the different dimensions of $\mathbf{z}_1$ and thus affects the metrics.

### 4.1 DISCUSSION

From Table 1 we see that OAT beats the previous baselines on the DCI metric while performing comparably on the other metrics. We noticed that in the DCI metric, OAT consistently got high scores for informativeness and disentanglement, but lower scores for completeness. The completeness score is high if each factor is completely captured by only a single latent variable. However, as pointed in Eastwood & Williams (2018) enforcing completeness might be counter-productive to learning disentangled representation for complicated factors like rotation.

With the latent traversals 4 we see that the factors are not confined to single latent dimensions but instead are encoded over several dimensions. For example, complicated factors like rotation and

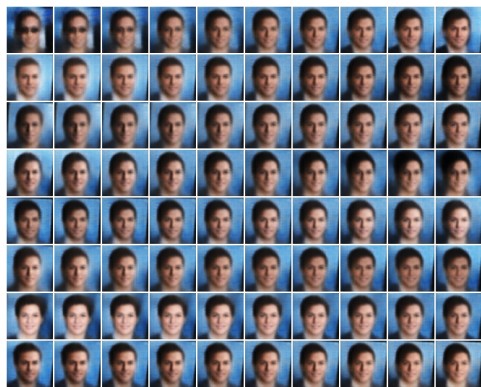 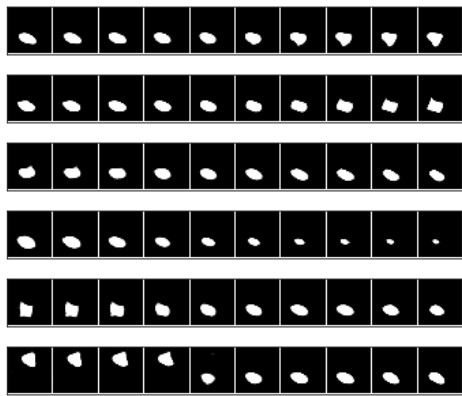

Figure 2: (Left): Traversals for the CelebA dataset show that OAT($z_1$=10,$z_2$=24) is able to disentangle multiple unique factors smoothly on real-world data, with unknown true factors. (Right): Results showing the OAT ($z_1$=5,$z_2$=10) architecture disentangling various factors for a synthetic dataset. From top to bottom: oval to heart, oval to square, rotation, and size. The rest of the values for different factors are encoded in different dimensions as completeness is not enforced during disentanglement

| Model | FactorVae | MIG | DCI |
|---|---|---|---|
| FactorVAE ($\gamma$=10) | 0.33 ± .06 | 0.14 ± .02 | 0.75 ± .01 |
| FactorVAE($\gamma$=40) | 0.42 ± .01 | 0.32 ± .03 | 0.73 ± .02 |
| $\beta$-TCVAE | 0.53 ± .01 | 0.38 ± .02 | 0.76 ± .01 |
| InfoGAN-CR | **0.68 ± .01** | 0.26 ± .03 | 0.53 ± .04 |
| OAT ($z_1$=10, $z_2$=10) | 0.64 ± .18 | **0.39 ± 0.04** | **0.77± .01** |

Table 2: Comparisons of the disentanglement metrics on the 3DShapes dataset averaged over 10 runs with different random seeds.

shape are encoded over two latent dimensions. This is a possible explanation for the lower MIG scores as the MIG score essentially computes compactness along with informativeness while giving less importance to modularity. As pointed in (Zaidi et al., 2021) MIG has a high score even if one latent dimension encodes information about multiple factors as long as one factor is only encoded by one dimension. This also explains why OAT($z_1$=5, $z_2$=10) has a higher MIG score than the other OAT models. However simpler factors like size are consistently encoded in a single latent variable.

From the ablation studies we see that when $|z_1|$ is large, some factors are encoded in more dimensions of $z_1$, whereas when $|z_1|$ is small, more than one factor is forced into single dimension of $z_1$. Thus, this model performs best when $|z_1|$ is larger than the true number of factors of variation if known.

We also notice that for some factors like y-axis, x-axis there is an abrupt change in the traversals. We believe this is an artifact of sampling interventions from the prior, and would change if a different intervention method is used. We use a zero mean unit variance Gaussian distribution for sampling. In future works we plan to experiment with sampling from different distribution like learned priors and approximate posterior distribution.

## 5 CONCLUSION

In this work we present a novel generative neural network framework for unsupervised disentanglement, One at a Time (OAT) Factor Learning. We demonstrate that with the use of unsupervised interventions, the network is able to learn smooth traversals across each latent dimension, and that the latent dimensions learned are informative, interpretable and separate without the use of labels. With the addition of the two separate latent spaces, OAT is able to learn an arbitrary number of factors, whereas before the latent dimension had to be pre-determined. Due to its design, it is able

to find a balance between the training stability of VAEs with the generative quality of GANs, while catering to real datasets with arbitrary number of factors. We hope that this work paves the path towards learning disentangled representations from real world datasets.

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

# A   APPENDIX

## A.1   INTERPRETABILITY OF DISENTANGLED REPRESENTATIONS

To be of use for downstream tasks, successful transfer learning and domain adaptation (Bengio, 2013) or to glean insights from observers be them machines or humans, the representations must somehow be interpretable. Interpretability has a simple intuitive understanding: that each factor represents a human-defined concept regarding the data, which humans could easily understand and identify. In practice, however, this is only intuitive because these are concepts pre-trained in human brains, and are not as clearly separable for machines without any imported biases; it is easy for the network to fuse multiple of what appear to humans as "essential factors" into one latent code while still keeping the factors independent and orthogonal to one another other; As an example, the network simply can learn a rotation of the "human" representation, which would appear from an observer to combine factors together. To address this, we define interpretability here in a new form not as uniquely human interpretable persay but as *interpretable latent representation manipulation*: each individual atomic code should make a unique and noticeable change in the output. Critically we are not defining noticable as being for humans only, as emphasized by the rotational example above, but by any observer.

## A.2   PROOFS

We assume a generative model $p_\theta(x, z) = p_\theta(x|z)p(z)$ where $p(z)$ is the prior assumed on the latent space and $p_\theta(x|z)$ is modelled by the generator. The corresponding inference model is $q_\phi(x, z) = q_\phi(z|x)q(x)$ where $q(x)$ governs the empirical data distribution of observed data while the inference model is $q_\phi(z|x)$.

The ELBO in [eq:1] can further be written as the KL divergence between the inference model and the generative model as follows:

$$
\begin{aligned}
\mathcal{L}_1 &= \mathbb{E}_{q(x)}[\mathbb{E}_{q_\phi(z|x)}[\log p_\theta(x|z)] - \mathrm{KL}(q_\phi(z|x)||p(z))] \\
&= \mathbb{E}_{q_\phi(x,z)}[\log p_\theta(x|z)] - \mathbb{E}_{q(x)}\mathrm{KL}(q_\phi(z|x)||p(z))] \\
&= \mathbb{E}_{q_\phi(x,z)}[\log p_\theta(x|z)] - \mathbb{E}_{q(x)}\mathbb{E}_{q_\phi(z|x)}\log\frac{q_\phi(z|x)}{p(z)} \\
&= \mathbb{E}_{q_\phi(x,z)}[\log p_\theta(x|z) - \log q_\phi(z|x) + \log p(z)] \\
&= \mathbb{E}_{q_\phi(x,z)}[\log p_\theta(x, z) - \log q_\phi(z|x)] \\
&= \mathbb{E}_{q_\phi(x,z)}[\log p_\theta(x, z) - \log q_\phi(z|x) + \log q(x) - \log q(x)] \\
&= \mathbb{E}_{q_\phi(x,z)}[\log p_\theta(x, z) - \log q_\phi(x, z)] + \mathbb{E}_{q_\phi(x,z)}\log q(x) \\
&= -\mathrm{KL}(q_\phi(x, z)||p_\theta(x, z)) - H(q(x)) \\
&\leq -\mathrm{KL}(q_\phi(x, z)||p_\theta(x, z))
\end{aligned}
$$

By maximizing the lower bound $\mathcal{L}$ we minimize the KL divergence, which is equivalent to the maximum likelihood objective as follows:

Minimizing the first term gives us the maximum likelihood objective. Given the asymmetric nature of the KL divergence this ensures that our generative model $p_\theta(x)$ has non zero probability for every $x \sim q(x)$. However, this does not ensure that the model gives a low probability to images not in $q(x)$. Thus we seek to also minimize the reverse KL divergence.

The ELBO in [eq:6] can further be written as the KL divergence between the generative model and the inference model as follows:

$$
\begin{aligned}
\mathcal{L}_2 &= \mathbb{E}_{p(z)}[\mathbb{E}_{p_\theta(x|z_t)}[\log q_\phi(z|x_t)] - \mathrm{KL}(p_\theta(x|z_t)||q(x))] \\
&= \mathbb{E}_{p_\theta(x,z)}[\log q_\phi(x|z)] - \mathbb{E}_{p(z)}\mathrm{KL}(p_\theta(x|z_t)||q(x))] \\
&= \mathbb{E}_{p_\theta(x,z)}[\log q_\phi(x|z)] - \mathbb{E}_{p(z)}\mathbb{E}_{p_\theta(x|z)}\log\frac{p_\theta(x|z_t)}{q(x)} \\
&= \mathbb{E}_{p_\theta(x,z)}[\log q_\phi(x|z) - \log p_\theta(x|z_t) + \log q(x)] \\
&= \mathbb{E}_{p_\theta(x,z)}[\log q_\phi(x,z) - \log p_\theta(x|z_t)] \\
&= \mathbb{E}_{p_\theta(x,z)}[\log q_\phi(x,z) - \log p_\theta(x,z_t) + \log p(z)] \\
&= -\mathrm{KL}(p_\theta(x,z)||q_\phi(x,z)) - H(p(z_t)) \\
&\le -\mathrm{KL}(p_\theta(x,z)||q_\phi(x,z))
\end{aligned}
$$

By minimizing the first term we ensure that the learned aggregated posterior $q_\phi(z)$ is similar to the factorized prior distribution. This is important for disentangling the factors, as we want the learned representation to be independent like the prior.

Thus we see that by maximizing both $\mathcal{L}_1 + \mathcal{L}_2$ we minimize the symmetric KL divergence.

### A.3 EXPERIMENTAL DETAILS

For our experiments we use the standard VAE architecture proposed in Kim & Mnih (2019) for the encoder and the decoder model. We use $[0, 1]$ normalised data as targets for the mean of a Bernoulli distribution $\log p_\theta(x|z)$. We use the negative cross entropy as the reconstruction loss and optimize the model using the Adam optimiser (Kingma & Ba, 2017) with a learning rate of $1e - 04$ for the dSprites dataset and $5e - 05$ for the 3D shapes dataset and the CelebA dataset.

For the discriminator we use the architecture proposed in Lin et al. (2020) and use the Adam optimizer with a learning rate of $1e - 04$ and $1e - 05$ for dSprites and 3Dshapes respectively with $\beta_1 = 0.9$ and $\beta_2 = 0.999$.

We train the models on the dSpirtes dataset for 30 epochs and on the 3DShapes and the CelebA dataset. for 50 epochs with a batch size of 128. To calculate the metrics we use 10 different random seeds and calculate the average of all the runs for the scores.

### A.4 FURTHER ABLATION RESULTS

We experimented with different dimensions of $\mathbf{z}_2$ and noticed that a larger $|\mathbf{z}_2|$ produces crisp reconstructions for datasets with a fixed number of factors of variation as can be seen in A.4 and A.4. However, the metrics do not change much as the dimension of $z_2$ is increased beyond 10.

For real datasets too we notice that a larger $z_2$ produces reconstructions with more details and more factors. This facilitates the discovery of more factors in the disentangled set during the OAT procedure as can be seen in **??**

### A.5 EVALUATION METRICS

For implementation details and hyperparameter settings of the metrics, we directly follow the settings in Locatello et al. (2019). Our VAE architecture is the one Kim & Mnih (2019) use in their experiments and the discriminator architecture is based on Lin et al. (2020).

#### A.5.1 BETAVAE, FACTORVAE

We evaluate the learned representations with one metric from each of the three kinds of metrics as described in (Zaidi et al., 2021). The intervention-based metrics compare representations by creating subsets of data in which one or more ground-truth factors are kept constant. These metrics do not make any assumptions on the factor-code relations which is their main advantage. We use the Factor-VAE metric from the intervention-based metrics kind. In this metric, in a batch a factor

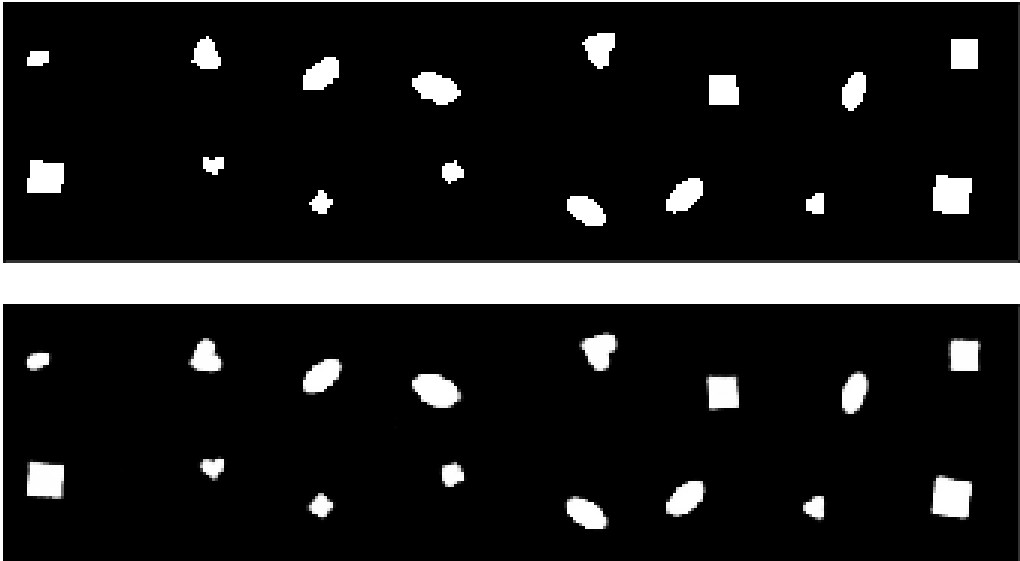

Figure 3: *Top:* The original input, *Bottom:* Reconstructions after passing through trained full OAT ($z_1$=10,$z_2$=10) model (as described in Experimental Setup.) Note that the model produces crisp, non-blurry, reconstructions.

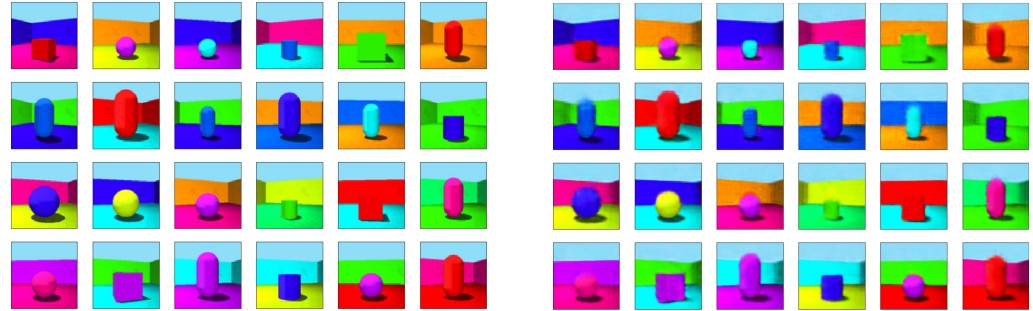

Figure 4: Training images (left) and the corresponding reconstructions (right) for $z_1$=10 and $z_2$=20.

$G_i$ is chosen randomly.Then, a fixed number of pairs from the data are selected where the value of the factor $G_i$ is the same. The intuition is that representation dimensions associated with the fixed factor should have the same value, which means a smaller difference than the other representation dimensions. Finally, a linear classifier is trained on the data set to predict which factor was fixed. The accuracy of the classifier is the score.

### A.5.2 DCI

Predictor-based metrics use regressors or classifiers to predict factors from the representations. These metrics train models to predict factor realizations from the representations. Then the usefulness of each code dimension in predicting a given factor is analyzed. These methods are naturally suited to measure explicitness. We use the DCI-Lasso and (Eastwood & Williams, 2018) metrics to measure explicitness and modularity.

### A.5.3 MIG

Information-based metrics compute a disentanglement score by estimating the mutual information (MI) between the factors and the representations. These methods require fewer hyper-parameters than intervention-based and predictor-based metrics. Moreover, they do not make assumptions on

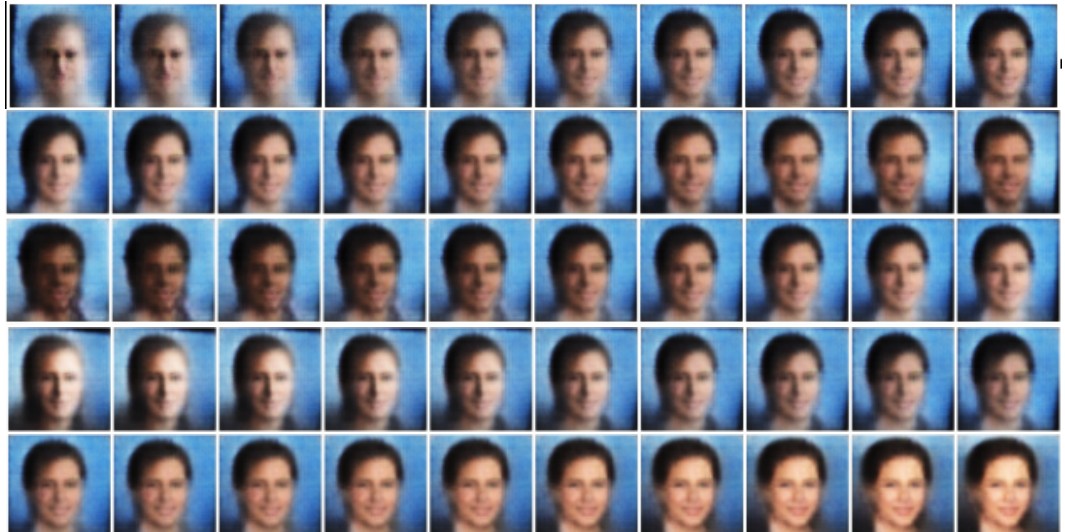

Figure 5: More Traversals for the CelebA dataset, for the same run (no cherry-picking between runs.) OAT is able to disentangle multiple unique factors smoothly on real-world data, with unknown true factors.

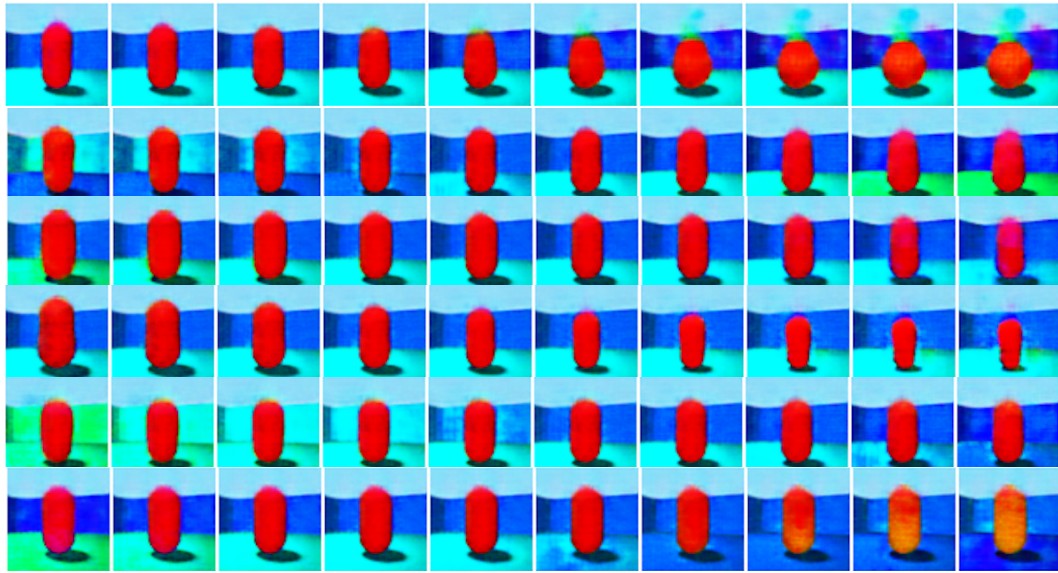

Figure 6: Traversals for the 3D shapes dataset with $|z_1|$=6 and $|z_2|$=10. Factors from top to bottom: shape, floor color, orientation, size, wall color, floor color.

the nature of the factor-representation relations. We use the MIG (Chen et al., 2016) to measure all the three facets of disentanglement.

### A.5.4  LATENT TRAVERSALS

For the CelebA dataset the ground truth values or the factors of variations are not known apriori. Therefore we use latent traversals 4 as a way to measure disentanglement qualitatively (Che et al., 2017). Disentanglement is evaluated qualitatively by traversing the latent space, by fixing all the dimensions of the representations except one and varying the values of that one dimension. For the varying dimension discrete values are sampled over its distribution and the resulting generated samples are visualized. A model has better disentanglement if for different values of the varying dimension, the resulting generated samples have a distinct and noticeable change for an interpretable factor of variation. We also qualitatively evaluate the models trained on dSprites and 3D shapes (ref: A.4).

### A.6  DOWNSTREAM CELEBA TASKS

To show that our method learns more informative factors than current state of the art approaches, we perform a number of downstream tasks on the learned disentangled representations. The CelebA dataset has 40 binary attributes for each image in the dataset. We extract the latent representations from the best disentangling models, namely, $\beta$-VAE, Factor-VAE and the $\beta$-TCVAE and train 40 SVMs to predict each of the binary attributes. We use 1000 (1k) and 10000 (10k) examples as a training set for these SVMs and evaluate it's performance on 2k examples in the test set. The results of the table 3 are calculated by taking the average of the accuracies of the 40 SVMs.

| Model | kernel:linear, 10k | kernel:linear, 1k | kernel:rbf, 10k | kernel:rbf, 1k |
|---|---|---|---|---|
| $\beta$-VAE | 0.64 | 0.62 | 0.69 | 0.66 |
| FactorVAE ($\gamma$=40) | 0.72 | 0.71 | 0.75 | 0.74 |
| $\beta$ -TCVAE | 0.74 | 0.74 | 0.79 | 0.77 |
| OAT ($z_1$=10, $z_2$=10) | **0.79** | **0.78** | **0.81** | **0.80** |

Table 3:  Average accuracy over 40 SVMs trained for each model representation, using different number of training examples.

Thus we see that the representations learned by OAT are more informative than the state of the art VAE-based approaches. We believe that OAT will perform better in the downstream tasks where certain factors might not be present in all the images but only in a subset of them.

