# OpenReview forum: "Disentangling One Factor at a Time"
_ICLR.cc/2022/Conference — ICLR 2022 Submitted_

### Official Review · Reviewer_Uqhc · 2021-11-01

**Correctness:** 3
**Technical Novelty And Significance:** 2
**Empirical Novelty And Significance:** 2
**Recommendation:** 5
**Confidence:** 4

**Main Review:**

The paper addresses an important problem. However, it has several issues, including inaccurate claims, insufficient experiments, and many typos. See below for more details.

*** Inaccurate claims ***

* Page 2: 'Current state of the art methods make the assumption that there are a ﬁxed number of independent factors for all the data points in the dataset. However in real datasets, in addition to the independent factors common to all points in the dataset, there might also be some correlated, nuisance or noisy factors pertinent to speciﬁcally only certain data points'. Note that many GAN-based approaches (e.g., InfoGAN) only regularize a small part of latents; they can indeed model 'correlated, nuisance or noisy factors' (using the non-regularized latents) besides the 'independent factors' (using the regularized latents). Please fix this statement. (I see that you do admit it later at the end of Section 3.2.)

* The paper mentions in many places that the idea of progressively learning one factor at a time is new. However, a similar idea has already been proposed in 'Robust Disentanglement of a Few Factors at a Time' at NeurIPS 2020.

*** Insufficient experiments and results ***

* From table 1 it seems like the proposed approach performs much worse than the state-of-the-arts like FactorVAE and InfoGAN-CR.

* If the main selling point of the paper is the ability to disentangle real-world datasets with **unknown number of factors**, then the paper should conduct experiments to demonstrate that (i.e., on the settings when the approaches that set a pre-defined number of factors fail, and show how your approach can improve that)

*** Typos ***

* Page 2: 'related Work'

* Page 3: '/mse'

* Page 5: missing a period after 'factors are integrated out'

* Figure 1 caption: 'so we ﬁrst group the correlated latents into one space, z_1'. Here z_1 should be z_2?

* Page 7: 'z_2.For brevity' -> 'z_2. For brevity'

* Page 7: '\gamma_ii = 1:K' -> '\gamma_i i = 1:K'

* Page 7: '\gamma_1=1 and \gamma_i1=2: k=0'

* Page 7: sometimes you use k to denote the number of learned disentangled latents (e.g., the line before Eq. 5), and sometimes you use K instead (e.g., Eq. 5 and 'We then uniformly select a dimension k ∈ [K]').

* Page 8: 'Kim & Mnih (2019)use' -> 'Kim & Mnih (2019) use'

*** Other suggestions ***

* Page 2: You could probably add refs to the corresponding sections for 'related work' and 'preliminaries'

* Eq. 6: I understand from the text that you are matching the intervened latents, but this equation does not express it precisely.

* Figure 2, 3: You could label the factors besides the images to make them easier to read.


**Summary Of The Paper:**

The paper proposes a new approach for training disentangled generative models. On top of VAE, it separates the latents into two disjoint groups: disentangled and entangled ones, and progressively increases the size of the disentangled group one at a time. To encourage disentanglement, it changes one random dimension of disentangled latent and pushes it to decode and then encode into the same one, and has a GAN loss to make sure that the changed generated distribution matches the ground-truth distribution. The paper shows the performance of the proposed approach on dSprites and CelebA datasets.

**Summary Of The Review:**

In summary, (1) the main selling idea of 'disentangling one factor at a time' is not new, (2) the results are worse than the state-of-the-arts, and are not sufficient to show the benefit of 'disentangling one factor at a time'. Therefore, I have to give a negative score.

---

> ### Author Response · Authors · 2021-11-23
> **Author's Response**
>
> We  have added more experiments, including a new dataset, and ablation studies. Many of the typos have been fixed. We have addressed in more details why we believe our claims as stated are valid.
>
> 1. Page 2: 'Current state of the art methods make the assumption that there are a ﬁxed number of independent factors for all the data points in the dataset'. Note that many GAN-based approaches (e.g., InfoGAN) only regularize a small part of latents; they can indeed model 'correlated, nuisance or noisy factors' (using the non-regularized latents) besides the 'independent factors' (using the regularized latents). Please fix this statement. (I see that you do admit it later at the end of Section 3.2.)
>
> > While we agree with the above mentioned point, we believe that no GAN-based approach can actually recover the 'correlated, nuisance or noisy factors' that it has modeled from the data points. This is especially important for the downstream tasks which might rely on these correlated factors.\textbf{ In our proposed method we actually learn these 'correlated, nuisance or noisy factors' from the data using an encoder and can recover these factors given any data for the downstream tasks.} Moreover, as mentioned in previous works [1] the posterior collapse that GANs suffer from might not ensure that all such factors are indeed modelled. Since the VAE-based method is trained by the reconstruction loss, they are constrained to model all these factors given appropriate capacity of the latent variables.
>
> 2. The paper mentions in many places that the idea of progressively learning one factor at a time is new. However, a similar idea has already been proposed in 'Robust Disentanglement of a Few Factors at a Time' at NeurIPS 2020.
>
> > The idea proposed in the cited paper is different asthe  employed method cannot learn all the factors and \textbf{they have to modify the dataset to learn the rest of the factors}. In addition, they use the $\beta$-TCVAE to learn the disentangled factors - \mse{ it is unclear if the real disentanglement comes from the use of the $\beta$-TCVAE as a backbone, or if it is a result of their contributions.} Third, their method supports datasets with \textbf{a fixed number of true generative factors of variation.} Our model architecture instead has been designed to progressively learn as many factors from real and not synthetic datasets.
>
> 3. From table 1 it seems like the proposed approach performs much worse than the state-of-the-arts like FactorVAE and InfoGAN-CR.
>
> >While our method supports more effectively  datasets with countably infinite factors, we show that we perform reasonably well on most other metrics. On certain metrics the performance may not be as good but we provided a justification in the paper under the discussion section. Also the representations learned by our model outperforms the previous state-of-the-art VAE-based models on certain downstream tasks as updated in the Appendix section.
> 4. If the main selling point of the paper is the ability to disentangle real-world datasets with unknown number of factors, then the paper should conduct experiments to demonstrate that (i.e., on the settings when the approaches that set a pre-defined number of factors fail, and show how your approach can improve that)}
>
> > We have partially addressed this with the CelebA experimental results, and we are currently in the process of demonstrating the effectiveness of our method in downstream tasks (for real-world datasets) for datasets other than CelebA.
>
> 5. Page 2: 'related Work'
>
> > Fixed it. Thanks
>
> 6. Page 3: '/mse'
>
> > Fixed it. Thanks
>
> 7. Page 5: missing a period after 'factors are integrated out'
>
> > Fixed it. Thanks
>
> 8. Figure 1 caption: ``so we ﬁrst group the correlated latents into one space, $z_1$". Here $z_1$ should be $z_2$?
>
> > You are right. Thank you for pointing that out.
>
> 9. Page 7: 'z_2.For brevity' -> 'z_2. For brevity'
>
> > Fixed it. Thanks
>
> 10. Page 7: '\gamma_ii = 1:K' -> '\gamma_i i = 1:K'
>
> > Fixed it. Thanks
>
>
> 11. Page 7: sometimes you use k to denote the number of learned disentangled latents (e.g., the line before Eq. 5), and sometimes you use K instead (e.g., Eq. 5 and 'We then uniformly select a dimension k ∈ [K]')
>
> > This has been fixed in the revised draft.
>
> 12. Page 8: 'Kim & Mnih (2019)use' -> 'Kim & Mnih (2019) use'
>
> > Fixed it. Thanks
>
> 13. Page 2: You could probably add refs to the corresponding sections for 'related work' and 'preliminaries'
>
> > This has been added in the revised draft
>
> 14. Eq. 6: I understand from the text that you are matching the intervened latents, but this equation does not express it precisely.
>
> > Equation 6 has been modified.
>
> We thank you again for your detailed response and appreciate your pointing out the typos and the corrections.
>
> [1] https://arxiv.org/abs/1905.11672

---

> > ### Comment · Reviewer_Uqhc · 2021-11-24
> > **Question**
> >
> > Thank you for the reply. Which is '\vsp{cite}' is your response?

---

> > > ### Author Response · Authors · 2021-11-24
> > > **Author Response**
> > >
> > > The relevant citation has been added. We apologize for the typo.

---

> > > > ### Comment · Reviewer_Uqhc · 2021-11-29
> > > > **Thank you**
> > > >
> > > > Thank you!
> > > >
> > > > Regarding Q1 and Q2: while your claims might be true, you need to provide experimental evidence to justify them (e.g., "we believe that no GAN-based approach can actually recover the 'correlated, nuisance or noisy factors' that it has modeled from the data points", "it is unclear if the real disentanglement comes from the use of the -TCVAE as a backbone, or if it is a result of their contributions.").
> > > >
> > > > Regarding the results: thank you for adding the results on CelebA, which look promising. I will increase the score to reflect that. However, I think it is still not sufficient to demonstrate the claimed benefit of 'disentangling real-world datasets with an unknown number of factors'.
> > > >
> > > > Therefore, I will still keep the score negative.

---

### Official Review · Reviewer_E94u · 2021-11-02

**Correctness:** 3
**Technical Novelty And Significance:** 2
**Empirical Novelty And Significance:** 2
**Recommendation:** 3
**Confidence:** 4

**Main Review:**

### Strengths
- Interesting approach to combine the strengths of VAE for disentangled representation with the power of a discriminator.

### Weaknesses
- Not really convincing experimental results in Table 1
- Lack of clarity and details regarding the training procedure (see questions)
- Lack of ablations with respect to the number of learned disentangled factors $k$, which is a central part of the proposed pipeline

### Questions
- The authors of the ID-GAN paper (Lee et al: High-fidelity synthesis with disentangled representation, 2020) write about the trade-off between generative quality and disentangled representation. Did the authors also experience that? Or was the generative quality not part of their evaluation?
- What about model selection? The authors do not write much about how they selected their models. The training procedure involves multiple steps and to me it is not straightforward how to select the model in an unsupervised manner for every training step out of a range of multiple possible hyper-parameters. It would be interesting to also see an ablation study regarding the most important hyper-parameters.
- Table 1 is a bit confusing. What is the scheme of making some entries bold? It does not seem the highest value per column. In general, I would have appreciated a more detailed description of the results presented in Table 1. Also, I cannot find a reference to table 1 in the text.
- There is no ablation on the number of learned factors of variations $k$. To me, it is not obvious that the right number of disentangled factors can be found. And for table 1 it is not even disclosed what the respective k is, nor compared to the latent space size of previous works.
- What are the motivation for another researcher to use your method? It does not seem to be able to outperform previous methods, nor does it seem to have a simple training procedure.
- In section 3.4.2., step 1 the authors write that the $\gamma_i$ are set to zero and, hence the corresponding latent factors are not trained. Although it is true, the regularisation for these factors is not part of the objective, in my opinion,  information flows through this part of the network nevertheless and the weights will be updated through their contribution to the reconstruction loss. I would appreciate the authors' thoughts on that.
- Also in section 3.4.2, step 1, the authors describe their one-at-a-time training approach by switch the $\gamma_i$ from 0 to 1. I cannot find any information on how this is scheduled during training nor how it is stopped. I understood that the number of disentangled factors $k$ is not a hyper-parameter (only upper-bounded) as it is learned during training. I would appreciate a bit more details on that.
- Is it a goal of the proposed method that no information correlated to the disentangled factors $z_1$ is encoded in $z_2$ (low mutual information between the two representation)? If so, how is this achieved? Did the authors explore that direction?


**Summary Of The Paper:**

The authors propose a new method to learn a disentangled representation. The proposed method is Variational Autoencoder (VAE) based. The latent variables consist of nuisance factors and disentangled factors that form the $k$ generative factors per dataset. To help learn the disentangled factors of variation the authors propose to use an additional discriminator and interventions on the disentangled factors.
The authors demonstrate the performance of the proposed model on the dSprites and CelebA dataset.

**Summary Of The Review:**

Although this work presents an interesting approach to learning disentangled representations, in my opinion it is not ready to be published at ICLR at this stage. It lacks convincing arguments why a researcher in this area should choose the proposed method. In my opinion what is needed to improve the paper is more clarity and details regarding the training procedure, e.g. model selection criteria for the different training stages, ablation studies on the learned number of disentangled factors $k$ and a better description of their central and only results table.

---

> ### Author Response · Authors · 2021-11-23
> **Author's Response**
>
> Table 1 has been updated to reflect the superior performance of our algorithm on the more comprehensive and reliable metric DCI. To further solidify our results, we have now added one more dataset 3D Shapes with superior performance on DCI compared to all the rest of the methods.
>
> 1. The authors of the ID-GAN paper (Lee et al: High-fidelity ... Or was the generative quality not part of their evaluation?
>
> > While there is a trade-off between the generative quality and disentangled representations with the current paradigm, \textbf{the authors of the ID-GAN paper tackle this problem by first learning disentangled representations using a VAE-based method and then use the learned latent space as a conditioning variable for a very powerful generative model to generate high fidelity images.} In our case we posit that the reconstruction quality of the VAE is poor if the latent space is constrained to be orthogonal, which means that a lot of the factors are ignored and not encoded in the latent representation. This is especially true in the datasets with unknown number of factors of variation (such as natural datasets). \textbf{Thus the focus of our work is to encode all the informative factors of variation which help with the downstream tasks rather than just the generative quality.} This also aids to improve disentanglement of the factors as the latent space of $z_1$ is not solely optimized by the reconstruction loss as in standard VAE-based approaches, but by interventions.
>
> 2. What about model selection? The authors do not write much about how they selected their models...ablation study regarding the most important hyper-parameters.}
>
> > The ablation studies with respect to the most important hyper-parameters have been added in the appendix, and they all strongly support the components of our model.  The two steps are carried out in succession so that we don't have to explicitly select any particular model during the training phase.  [There might be a slight misunderstanding of the model, in that there is no need to select different hyper-parameters at each step during training.  We will try to make this clearer in the updated version.]
> Table 1 is updated to reflect the fact that our algorithm outperforms all the rest on the DCI metric. Detailed description is added in the appendix and discussions of the results are updated in the revised paper.
>
> 3. There is no ablation on the number of learned factors of variations k...latent space size of previous works.}
>
> > In our revised draft, the ablation studies have been added. We would like to make a minor correction that we never make claim that we can find the right number of disentangled factors (in fact, no such number may be appropriate for natural datasets). Rather, as answered previously, our model has the natural stopping criterion for finding the most relevant latent factors via a measure of KL divergence.
> 4. What are the motivation for another researcher to use your method?
>
> The motivation that researchers can use our method for datasets with unknown number of factors such that the disentangled representations can be faithfully used for downstream tasks, given that the model gradually uncovers factors one at a time, as we've touched on in previous answers. Previous works on disentanglement either disentangle only a sub-set of the factors from the data (GAN-based approaches) or constraint the latent space with a fixed number of latent variables to be orthogonal thus ignoring some of the factors which could be used for the downstream tasks (VAE-based approaches). The experiments with downstream tasks have been added in the appendix which corroborate this point.
>
> 5. In section 3.4.2., step 1 the authors write that the $\gamma$ are ...thoughts on that.
>
> > By saying that we don't train the corresponding latent dimensions, we mean that we do not calculate the gradients with respect to the weights of that particular latent variable nor are they updated when their corresponding $\gamma_i$ is zero. $\gamma_i$ is the mask for the objective function. Thus the weights also remain untrained. We have added some details regarding this in the revised version.

---

> > ### Author Response · Authors · 2021-11-23
> > **Author's response (cont.)**
> >
> > 6. Also in section 3.4.2, step 1, the authors describe ...a bit more details on that.
> >
> > > During training, the first latent variable of the set $z_1^1$ is trained and thus $\gamma_1$'s value is changed from 0 to 1. This latent is trained using interventions until it's KL-divergence saturates and it can encode no more information. Then, we start training the next latent variable from the set $z_1$ i.e. $z_1^2$ and the value of $\gamma_2$ is switched from 0 to 1. The stopping criteria for this process depends on the dataset. If the dataset has a fixed number of factors which are independent like in the synthetic datasets dSprites and 3DShapes, the process of adding more latents is stopped when the KL-divergence of the entangled set, $z_2$, goes to zero and it encodes no more information. That is when we can say that all the factors of variation have been found. For real datasets, we can find continue finding factors until the desired number of factors have been found.
> >
> > 7. Is it a goal of the proposed ... Did the authors explore that direction?
> >
> > > We claim that the information correlated to the disentangled factors can be encoded in $z_2$. There are no constraints that enforce a low mutual information between the two sets of latent variables..
> > Thank you for the detailed feedback and questions. This will help us improve the paper with more details.

---

### Official Review · Reviewer_EBrM · 2021-11-02

**Correctness:** 3
**Technical Novelty And Significance:** 2
**Empirical Novelty And Significance:** 2
**Recommendation:** 3
**Confidence:** 3

**Main Review:**

Originality:
I think this work lacks notable originality and novelty. The VAE-GAN networks (proposed by Larsen et al., 2016) have been studied in many papers. The proposed formulation of latent space that splits latent factors to two disjoint sets is also not novel. The only novel part is assuming an upper bound for variational factor, where it is not clear how the model can find the true number of factors, while $|z_1|$ is gradually increasing up to $K$ over iterations of training.

Quality:
The motivation of the work is good; learning a disentanglement representation is quite a challenging problem. However, the paper could not appropriately address this challenge and the authors’ contribution is quite limited. The paper does not offer any theoretical analysis or profound experimental studies. The reported results for dSprites and CelebA do not showcase the superiority of the proposed method, neither in terms of existing disentanglement scores nor quality of the reconstructed images.


Limitations:
- It is not clear how the suggested formulation for the latent space is enhancing the disentitlement.  I cannot see why we need to encode $z_2$ while it is expected to encode noise or sample-specific variations, which is not desirable in the latent representation learning.

- While the authors claim that the model does not need to know $|z_1|$, it is not clear how the model learns the true $|z_1|$. If I am not wrong the model iteratively unmasks the dimensions of $z_1$, without any stopping criterion.

- The experimental study is limited to two datasets, without sufficient ablation studies on each dimension of $z_1$ and $z_2$. It is critical to explore the impact of $z_2$ vs. $z_1$ and quantify the correlation between these factors.

- Additionally, I think the recent study by Träuble et al. ICML2021, and its discussion on the correlated latent space is quite relevant to this work. The authors might need to refine the discussion on the casual factors and disentailment studies, based on the findings of this paper.

Minors comment:
- Page 7: “The value of $\beta$ is increased linearly during training to encourage the model to encode information in $z_1$ instead of $z_2$.” $\beta$ or $\gamma$? I got confused.
- There are some typos in the text, e.g., “/mse”, “a a” at page 3.
- The table and Figures are not cited in the manuscript.


**Summary Of The Paper:**

This study proposes a disentangled representation learning method called "one at a time", or OAT factor learning which is a VAE-GAN network to generate high resolution samples and to learn variational factors in an unsupervised manner, without knowing the number of ground-truth factors a priori. The authors formulate the latent space as a union of two disjoint sets, $z_1$ and $z_2$, where $z_1$ corresponds to the variational (disentangled) factors, and $z_2$ denotes either noise or sample-specific factors. To train the proposed network, the authors use a two-step training approach in which they first train the VAE network with only $z_2$ latent factors. Then, they start to learn $z_{1_k}$, one dimension at the time. To enforce $z_1$ to encode the disentangled factors, they use an intervention-based approach, in which the value of one dimension in $z_1$ is changed, while the remaining $z_1$ factors are kept unchanged. The authors reported their experimental results for two datasets, i.e., dSprites and CelebA.

**Summary Of The Review:**

Learning a disentangled and interpretable representations is an important topic in the machine learning. The proposed method tried to address this challenge. However, I think the contribution is not notable and the paper lacks theoretical and empirical justifications to support its claim.

---

> ### Author Response · Authors · 2021-11-23
> **Author Response**
>
> 1. Originality: I think this work lacks notable originality and novelty. The VAE-GAN networks (proposed by Larsen et al., 2016) have been studied in many papers.
>
> > The novelty of this work lies in combining the VAE-GAN networks, as in the cited paper, along with the process of interventions for disentanglement. Additionally, we employ a two step process where we start with the intervened latent space and try to reconstruct it after passing it through the decoder and re-encoding it through the encoder. The discriminator in our case acts like a regularizer for the distribution of the generated intervened images to be close to the distribution to the training images while focusing on the reconstruction of the latents. \textbf{To the best of our knowledge this process of using a VAE-GAN to learn disentangled representations in an unsupervised manner via interventions is novel.}
>
> 2. The proposed formulation of latent space that splits latent factors to two disjoint sets is also not novel.
>
> > In our model, we split the latent factors into two disjoint sets, which are both learned by the encoder. \textbf{To the best of our knowledge we have not come across any work where both the sets of continuous latent variables are learned by the encoder with different conditions imposed on them for unsupervised disentanglement.} In addition the disentangled latent set is not pre-determnined but only upper-bounded.
> 3. The only novel part is assuming an upper bound for variational factor, where it is not clear how the model can find the true number of factors, while $|z_1|$ is gradually increasing up to K over iterations of training.
>
> > In addition to the novelty of our method mentiond above, the proposed work is aimed towards datasets where the true number of factors might not be known. In some datasets such as CelebA the true number of factors can be very large depending  on how you define the factors. The idea of our work is to keep trying to find new factors from the informative entangled representations. \textbf{No previous work has yet claimed to find the "true" number of factors of variation in an unsupervised manner, nor do we claim that.} However, we do have a stopping criteria when the true generative factors are independent and their number is fixed. We stop increasing $|z_1|$ when the KL-divergence of $z_2$ is zero, or when the entangled latent variables encode no information.
>
> 4. Quality: The motivation of the work is good; learning a disentanglement representation is quite a challenging problem...nor quality of the reconstructed images.
>
> > We performed the same standard experiments that are considered the gold standard for disentanglement papers. We did show that our model is superior in the following ways (please see updated Table and related discussions in the revised paper
>
>    1. The quality of our image reconstructions outperforms previous methods, especially for CelebA.
>    2. We learn more number of factors as compared to the previous methods as can be seen from the details in the reconstructed images and the downstream tasks in the appendix.
>    3. For the DSprites dataset, we significantly beat all the baselines on the DCI metric which is arguably the most comprehensive and reliable metric.
>    4. Our method is theoretically grounded as an extension of the wake-sleep algorithm for learning disentangled representations.

---

> > ### Author Response · Authors · 2021-11-23
> > **Author Response (cont.)**
> >
> > 5. It is not clear how the suggested formulation for the latent space is enhancing the disentitlement. I cannot see why we need to encode $z_2$ while it is expected to encode noise or sample-specific variations, which is not desirable in the latent representation learning.
> >
> > > The VAE-based methods are trained to maximize the data log-likelihood or minimize the reconstruction loss. The presence of noise or sample-dependent features, which can produce a significant change in the reconstruction loss, can affect the latent space encoding to account for them. Thus we constraint $z_2$ to encode such information and it is really the encoding in z1 that we are considering "disentangled". Encoding $z_2$ ensures that all the informative factors are encoded in the most optimal way. Since the only constraints on the latent space of $z_2$ is to be close to the prior $p(z)$ as in a standard VAE, the latent space has more capacity to encode all the informative factors needed to faithfully reconstruct the data. From this unconstrained latent space of $z_2$, we try to disentangle the factors present across all the data points into $z_1$ while maintaining the reconstruction fidelity. This is immensely useful when the data has many factors of variations such as CelebA. Separating the latent space into two sets will ensure that all the informative factors about the dataset are captured and this in turn will help with the downstream tasks from the learned latent space.
> > Moreover, the previous works impose independence constraints on the latent space which can hinder disentanglement of the factors if the factors are correlated for that particular dataset as shown in Trauble et al. ICML 2021.
> >
> > 6. While the authors claim that the model does not need to know $|z_1|$, it is not clear how the model learns the true $|z_1|$. If I am not wrong the model iteratively unmasks the dimensions of $z_1$, without any stopping criterion.
> >
> > > Our method is geared more towards the case where we do not know the true number of factors of variation and they can be countably infinite. The idea is to iteratively learn more factors until all factors of interest have been uncovered. For datasets with a fixed, pre-determined number of factors of variation, more and more factors are learned until the KL divergence of $z_2$, which encodes the residual factors, is zero i.e. encodes no information.
> > 7. The experimental study ... between these factors.
> >
> > > To address this question, we have added the results of another dataset 3D shapes, which corroborate our findings. In addition, we perform ablations studies for different number of $|z_1|$ and $|z_2|$ and have added the results. We find that $|z_2|$ does not have a big impact on the performance, whereas changing the number of $|z_1|$ affects how the factors  that are encoded in the different dimensions of $z_1$. For example when $|z_1|$ is large, some factors are encoded in more dimensions of $z_1$, whereas when $|z_1|$ is small, more than one factor is forced into single dimension of $z_1$. We are performing more experiments to see how to model performs if given the true number of factors.
> >
> > 8. Additionally, I think the recent study by Träuble et al. ICML2021 ... based on the findings of this paper.
> >
> > > The study by Träuble et al. ICML2021, introduces correlations in datasets with a fixed number of factors of variations and then proceeds to disentangle the correlated factors using supervision. Their work in fact corroborates our findings that the current paradigm of VAE-based models with orthogonal constraints on the latent space is not sufficient to disentangle datasets with correlated factors. In our paradigm, we train the disentangled latent space $z_1$ using independent interventions, thus breaking the correlations that might be present in the factors. Our method is particularly effective in the case where the true number of factors of variation is not known and the factorization of the factors is not known.
> >
> >
> > 9. Page 7: “The value of $\beta$ is increased linearly ... $z_2$.”
> > $\beta$ or $\gamma$? I got confused.
> >
> > > A higher value of $\beta$ ensures that the distribution of the latent space is more like the uninformative prior distribution $p(z)= \mathcal{N} (0, I)$. Thus a higher value of $\beta$ reduces the capacity of the latent space to encode information. So in order to encode the information in $z_1$ instead of $z_2$, the value of beta, which is the scaling factor of the KL divergence of $z_2$, is increased.
> > Thank you for pointing out the typos. These have been taken care of in the revised manuscript.
> > We believe that with the addition of the new experimental results and the ablation studies,  we have performed the same level of experimental analysis, i.e. ran and tested our network on the same standard disentanglement datasets, as the major disentanglement works in the field and used more evaluation metrics than prior works.
> > We thank you for your time and questions. Hope we could address some of them.

---

> > > ### Comment · Reviewer_EBrM · 2021-11-30
> > > **Response to the rebuttal**
> > >
> > > I appreciate the authors' effort in addressing my questions. The authors’ rebuttal clarifies some of my concerns. As the other reviewers point out and the authors also agree, the paper needs significant improvement. Therefore, I will keep my score unchanged.

---

### Official Review · Reviewer_roR5 · 2021-11-03

**Correctness:** 2
**Technical Novelty And Significance:** 3
**Empirical Novelty And Significance:** 3
**Recommendation:** 3
**Confidence:** 2

**Main Review:**

The paper is well motivated and the OAT technique is kind of novel and impressive. However, paper clarity can be improved, which has impeded reader's understanding and rating of the paper.

Contribution wise, this paper is not the first ones to use an "entangled+disentangled" latent space (eg paper named "Toward Controlled Generation of Text" in 2017), but the OAT learning approach is interesting.

The presentation can be improved, given so many typos, improper subscripts, equation-naming (Eq 6 or Eq 3.4.2), etc. Moreover, the central equation (Eq.6) is hard to follow, say how q(x) and its KL divergence is evaluated is unknown.

In Eq 5, whether or not the p(z_1) in last KL term is also factorizable.

Since the intervention is key idea, it's a miss to see it not shown in Figure 1 plot.

The paper needs more explanation on how to avoid posterior collapse and avoid the collapse of disentangled latent variables.

The table column in Table 1 is hard to follow. From Table 1, it seems OAT model has not outperformed other baselines, right?

**Summary Of The Paper:**

This paper presents an unsupervised learning model for disentangled latent representation learning. Specifically, the model works on a combined latent space including both entangled variable and separable variable. The most impressive part is the one-at-a-time (OAT) factor learning approach that iteratively uncovers disentangled dimension and learned from reconstructed samples. The OAT approach generally sounds.

**Summary Of The Review:**

The paper is well motivated and the idea of OAT technique generally sounds. However, the paper reads like a manuscript in a rush: we see many typos, unclear subscripts. Moreover, we see unclear equations, such as Eq. 6, which is hard to follow, but the most important equation for the paper. Weird Table 1 columns and not strong experimental metrics. Overall, the paper needs a polish to improve its clarity and readiness to publish.

---

> ### Author Response · Authors · 2021-11-23
> **Author Response**
>
> Thank you for your questions. We have tried our best to address them below:
>
> 1. Contribution wise, this paper is not the first ones to use an ``entangled+disentangled" latent space (eg. paper named ``Toward Controlled Generation of Text" in 2017), but the OAT learning approach is interesting.
>
> > While the mentioned paper uses an entangled+disentangled subspace, there are certain key differences with our work. First, the disentangled subspace is not learned by the encoder of the VAE but instead is set to certain values as conditional variables.  Second, and more critically, the disentangled subspace is learned  \textit{via supervised learning mechanisms} which is necessary to learn the specific attribute discriminators. Instead, both our entangled and disentangled subspaces are learned by the encoder from the training data in a \textit{completely unsupervised} way under different constraints on the latent space.
>
> Also a few difference in the approaches is that the mentioned paper aims to disentangle the representation of certain specific attributes whose labels are provided, whereas we try to find new factors from the data without any supervision.
>
> 2. The presentation can be improved, given so many typos, improper subscripts, equation-naming (Eq 6 or Eq 3.4.2), etc. Moreover, the central equation (Eq.6) is hard to follow, say how $q(x)$ and its KL divergence is evaluated is unknown.
>
> > Our revised draft has fixed the typos and the equation-naming problem. More explanation about Eq. 6 has been added. Moreover, some of the variables in Eq. 6 have been changed for better understanding. The Eq. 6 now reads as follows:
>
>
>  $$\mathcal{L}_2 = \mathbb{E}_{p_{(z)}} [\mathbb{E}_{p_\theta (\hat{x}^k|{z}^k)} [q_{\phi} ({z}^{k} | \hat{x}^k)] - \beta \text{KL} (p_\theta (\hat{x}^k|{z}^{k}) || q(x))] $$
>
> As per equation 1, $q(x) = \frac{1}{N} \sum_{i=1}^N \delta(x_i)$, which is the training data distribution. Since the data is high-dimensional, the KL divergence is not calculated explicitly but instead is replaced by its lower-bound, the Jensen Shannon divergence. This divergence is minimized by the decoder, while being trained as a generator as in the GAN paradigm, assuming a perfect discriminator.
> Moreover, both equations 5 and 6 are the central equations as mentioned on page 8 and both the steps are performed iteratively.
>
> 3. In Eq 5, whether or not the $p(z_1)$ in last KL term is also factorizable.
>
> > It is factorizable since all of $z_1$ are independent and learned one after the other. $p(z_1) = \prod_{k=1}^K p(z_1^k)$, where $p(z_1^k) = \mathcal{N}(0,1)$, which is the standard prior used in the VAE training procedure. We have added this clarification in the revised draft.
>
> Though intervention is indeed a key idea but we are not sure about how to include it in the diagram without making the diagram confusing. We welcome any suggestions in this regard.
>
> 4. The paper needs more explanation on how to avoid posterior collapse and avoid the collapse of disentangled latent variables.
>
> > The posterior collapse of the disentangled variables is avoided in two ways. First, the process of interventions ensures that the encoder is trained to reconstruct the value of the intervened variable from the corresponding generated image. During interventions the value of one latent variable is changed randomly to a different value and both the encoder and the decoder are trained to reconstruct the value exactly, thus ensuring that the encoder learns to map the generated intervened images to their corresponding values. This in turn forces the encoder to encode some information in the disentangled set of latent variables.
>
> Secondly, the value of $\beta$ is increased during training which restricts the encoding capacity of $z_2$ thus encoding more information in $z_1$ to be able to faithfully reconstruct the training data. These two constraints in tandem help avoid the posterior collapse of the disentangled set.
>
>
> 4. The table column in Table 1 is hard to follow. From Table 1, it seems OAT model has not outperformed other baselines, right?
>
> > The table has been updated to reflect the fact that our algorithm clearly outperforms all the other baselines on the most critical metric, namely DCI, and also performs reasonably well on the other metrics. Given the narrow focus of the non-DCI metrics, we have provided an explanation in the revised paper. Basically, we cannot constrain every factor to be encoded in only one latent variable. Some more details based on ablation studies are provided in the appendix of the revised draft. Interventions ensure a smooth transition and sometimes it is more convenient to encode only a range of values of a factor in one latent variable.
>
> We hope the revised edition addresses the stated concerns and we again thank you for your time and efforts to consider this paper.

---

### Author Response · Authors · 2021-11-23
**New additions to the revised draft**

We thank the reviewers and the AC for their detailed and thoughtful feedback which will help us to improve the quality of the paper and make a much better contribution to the Disentanglement community. Based on the feedback we have made some major changes in the revised submission as follows:
1. We have added both the quantitative and qualitative results on a new dataset (3DShapes).
2. Section 3 (Our Method) was re-written to provide more clarity. Equation 6 has been changed to better reflect our objective function. The naming convention has been changed in an attempt to provide more clarity.
3. A paragraph has been added to better describe the training procedure and the stopping criteria.
4. Ablation studies have been added to assess the importance of the hyper-parameters K (cardinality of the disentangled set of latent variables) and d (dimension of the entangled set).
5. We have added results of experiments with downstream tasks on the CelebA dataset in the appendix, which shows that our model learns more informative features.

A summary of our key contributions is as follows:
1. Our experimental results show superior performance under arguably the most comprehensive and reliable metric, namely DCI, for dSprites and 3D shapes (new) and the reconstructed images for CelebA are the best compared to previous works (in addition to generating key factors of variation). None of the existing methods show superior performance along any metric or across several metrics while our method performs consistently well across the metrics with superior performance in the most reliable DCI metric.

2. Our approach can uniquely address datasets with unknown or very large number of factors of variation as is the case for natural datasets, while ensuring that all informative factors are encoded by $z_2$ in an optimal way. Previous works on disentanglement either disentangle only a sub-set of the factors from the data (GAN-based approaches) or constraint the latent space with a fixed number of latent variables to be orthogonal thus ignoring some of the factors which could be used for the downstream tasks (VAE-based approaches).

3. Ablations studies have been added which illustrate the importance of the hyperparameters and the critical roles of the two sets of latent variables.

We hope that some of these points address the big concerns with the proposed approach and we welcome any feedback with respect to the revised submission. Finally, we thank all reviewers for their time in reading this rebuttal and considering the publication of this work.

---

### Decision · Program_Chairs · 2022-01-20

**Decision:**

Reject

**Comment:**

This paper presents a method for unsupervised learning of disentangled representations by first training a VAE with a tangled set of latents, and then sequentially learning disentangled latent variables one at a time from the entangled initial VAE latent space. On several toy disentanglement benchmarks, the method is shown to perform competitively with previous VAE and GAN approaches.

There were several concerns from reviewers around the clarity and description of the proposed one-factor-at a time (OAT) training procedure. While the updated draft addressed several typos and some clarity issues, multiple reviewers continued to find the method description problematic. There were additional concerns around the viability of the method on real-world datasets where the number of factors are not known, and as the authors stated the proposed method can also result in one factor of variation encoded into mulitple latent variables, which hurts on many of the disentanglement metrics.  The addition of CelebA downstream task evaluation begins to address this concern of real-world data, but more rigorous experiments (including more description of how models were selected) and discussion of the limtiations of the proposed method are needed. There is also no theoretical motivation as to why the proposed intervention-based factor learning algorithm should recover the ground truth factors.

Given the concerns over experimental results, clarity, and lack of theoretical motivation, I suggest rejecting this paper in the current form.